# Dephasing enhanced transport of spin excitations in a two dimensional lossy lattice

**Andrei Skalkin** [1], **Razmik Unanyan**[1], **Michael Fleischhauer**[1*]

**1** Department of Physics and Research Center OPTIMAS, University of
Kaiserslautern-Landau, 67663 Kaiserslautern, Germany
⋆ mfleisch@rhrk.uni-kl.de

## Abstract

Noise is commonly regarded as an adverse effect disrupting communication and coherent transport processes or limiting their efficiency. However, as has been shown for example for small light-harvesting protein complexes decoherence processes can play a significant role in facilitating transport processes, a phenomenon termed environment-assisted quantum transport (ENAQT). We here study numerically and analytically how dephasing noise improves the efficiency of spin excitation transport in a two dimensional lattice with small homogeneous losses. In particular we investigate the efficiency and time of excitation transfer from a random initial site to a specific target site and show that for system sizes below a characteristic scale it can be substantially enhanced by adding small dephasing noise. We derive approximate analytic expressions for the efficiency which become rather accurate in the two limits of small (coherent regime) and large noise (Zeno regime) and give a very good overall estimate. These analytic expressions provide a quantitative description of ENAQT in spatially extended systems and allow to derive conditions for its existence.

# 1   Introduction

Understanding transport phenomena plays a central role in fundamental physics as well as for applications ranging from information technology to biology. Moreover their investigation can provide insightful information about key features of the system under investigation. Transport processes become especially rich on the nano-scale where quantum and classical effects compete. Of particular interest here is the so-called environment-assisted quantum transport (ENAQT) which has drawn attention due to the recent advancements in experimental analysis of exciton transport in biological systems, with the Fenna-Matthews-Olson protein complex (FMO complex), present in green sulfur bacteria, being a prime example. The FMO complex consists of several coupled molecules and serves as a transport network for excitons between a light harvesting antennae, where a sunlight photon absorption event creates an electronic excitation, and a reaction center, where the exciton energy eventually drives chemical reactions [1]. The remarkable feature of the FMO complex is its near-unity efficiency [2]. Using modern spectroscopic techniques it has been demonstrated that coherence between distant molecular aggregates persist throughout the entire exciton transfer process in the FMO complex despite being subjected to noise [3] [4]. Since then a discussion about the pivotal role of dephasing in assisting quantum transport has been raised [5]. These findings motivated numerous experiments focused on the simulation of noisy transport which were carried out in different artificial frameworks such as trapped ions [6], optical waveguides [7] [8], coupled electronic oscillators [9], and superconducting circuits [10].

The theoretical research of non-equilibrium ENAQT has been mainly conducted for the cases of a dimeric model describing a pigment-protein complex (i.e. minimal two-site system) taking into account non-Markovian effects [11], few-site models of FMO complex [12–14] or binary trees [15] for the single-excitation regime, all showing that dephasing can significantly improve transport compared to the noiseless case. By a specific example of fully connected networks (coupling between any two sites is identical) a concrete mechanism behind the phenomenon has been proposed using the notion of an invariant subspace, spanned by the states which have no overlap with the target site, in close analogy to dark states. Noise breaks the invariant space and more states contribute to the transport [13].

Because of its relevance for the collection of excitations in antenna systems such as light-harvesting complexes and in prospect of the rapidly developing experimental techniques with cold atoms [16–19], we here study transport of a single spin excitation from a random initial site in a lossy, 2D lattice model, see Fig. 1a. The quantities of interest for us are (i) the asymptotic efficiency $\eta$ of the extraction of this excitation at a specific collection site and (ii) the time $\tau$ this process takes. We study how the transfer depends on the microscopic parameters of the different unitary and non-unitary processes and analyze in particular how it is affected by dephasing noise. In addition to the parameters characterizing the microscopic dynamics, the transfer efficiency depends also on the structure of the network along which the transport happens. Here we investigate a two-dimensional lattice with short-range transport, isotropic

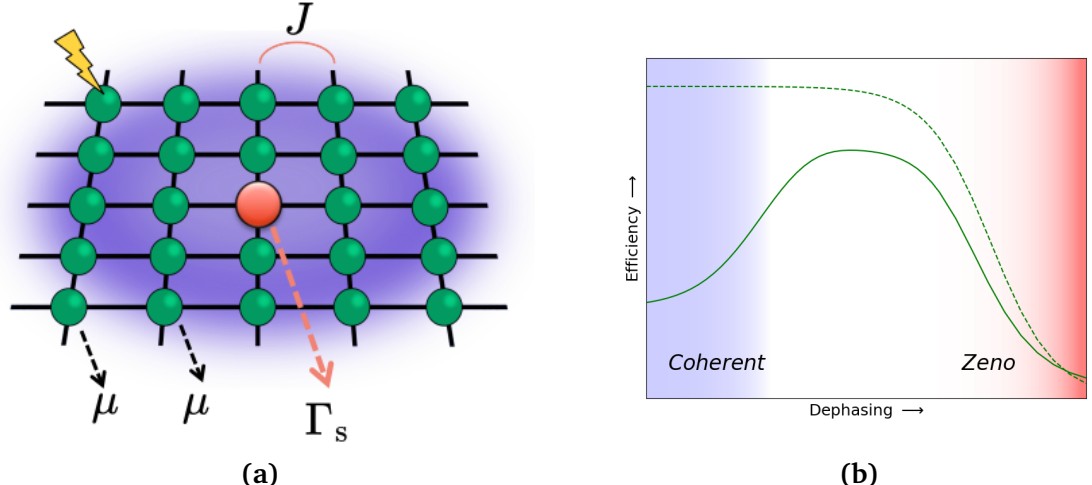

Figure 1: (a) Sketch of the system. Initially a spin up state is excited at a random site, starting to propagate in the lattice (with coupling $J$ between neighboring sites). This process is infringed by the spontaneous decay with rate $\mu$, while the favorable event of capturing at the central node is modeled as absorption with rate $\Gamma_s$. (b) ENAQT concept: while it is commonly expected that noise impedes transport (dashed line), it may improve transport efficiency for moderate noise strength (solid line).

in the two directions. Other network topologies will result in qualitatively different behavior. E.g. it is well known that a random walk in a 2D system will reach any site after infinite time. The same is however not true for the random walk in 3D, where the probability for this to happen is smaller than one, so even without background decay the transfer efficiency will be less than unity in a 3D lattice system. On the other hand, transport along tree-like networks can be faster, but requires that the extraction site is fixed.

We find for a 2D lattice that for linear system dimensions smaller than a characteristic length small dephasing indeed can lead to an significant increase of the transfer efficiency as schematically indicated in Fig. 1b. As in the studies of FMO complexes, we identify two regimes, a "coherent" regime where the efficiency increases with growing dephasing rate, eventually crossing over into a second, "Zeno" regime, where the transport gets increasingly suppressed. The non-monotonous behavior is the main feature of ENAQT, which we will study here in detail by numerical and analytic means. We establish analytic boundaries for the transfer efficiency in the two limits of low and high dephasing, assess optimal transport parameters and compare different regimes via numerical simulations. Compared to earlier work [15] we extend the system dimensionality to the 2D case with dipolar hopping and provide analytical insights into the transport process. We show that the non-monotonic behavior of the efficiency, characteristic for ENAQT, disappears if the system size becomes larger than some critical value determined by the ratio of the extraction rate to the rate of background absorption.

## 2   Model

We consider a quadratic lattice with $L^2$ sites and open or periodic boundary conditions, as sketched in Fig. 1a. At each site resides a two-level system with a ground and an excited state. Transport of excitations occurs between sites via an $xy$-type spin exchange process with an amplitude that decreases as the distance between the sites grows. The extraction of excitation

at the target site is described by a local decay with rate $\Gamma_s$. The transport of excitations from a random initial site to the target site and its subsequent extraction usually competes with unavoidable loss processes, which is here modeled by small local decay processes at each lattice site from the excited to the ground state with uniform decay rate $\mu$. Thus as the excitations hops on the lattice it may either decay spontaneously at any site or eventually be captured at the target center. Furthermore, external uncorrelated dephasing noise acts with uniform rate $\gamma$ at any position, destroying phase coherence between different paths (Fig. 1a).

In order to formalize the model we introduce the localised exciton states basis $\{|\downarrow_i\rangle, |\uparrow_i\rangle\}$ corresponding to eigenvalues $\{-1, +1\}$ of $\sigma^z$ and representing ground and excited states of the system at the position labelled by $i$. We restrict the model to a single up-state manifold, omitting interaction terms of $\sigma_j^z \sigma_i^z$-type in the Hamiltonian. Thus the Hamiltonian includes only spin hopping terms:

$$H = \sum_{i \neq j} J_{ij} \sigma_i^+ \sigma_j^- \tag{1}$$

where $J_{ij}$ are the corresponding coupling amplitudes ($J_{ij} = J_{ji}^*$), and $\sigma_i^+$ is the (spin-up flip) operator at site $i$. Having experimental realizations with Rydberg atoms in mind we assumed in the numerical simulations dipole-dipole interactions, though the analytical results are generalized for a wide class of Hamiltonians. For the dipole-dipole interaction one has:

$$J_{ij} = \frac{C}{\left|\vec{r}_i - \vec{r}_j\right|^\alpha} \tag{2}$$

where $C$ is the (dipole) coupling constant, $\vec{r}_i$ is the position of lattice site with index $i = 1, \ldots, L^2$, $L$ being the number of sites in each direction. We fix $\alpha = 3$ as a typical case. For a rectangular lattice with a lattice constant $a$ the tunneling rate between adjacent sites $J = C_3/a^3$ sets the time scale of transport.

The dynamics of the system is described by the Lindblad master equation ($\hbar = 1$):

$$\dot{\rho} = -i[H, \rho] + \mathcal{L}_{\text{diss}}(\rho) + \mathcal{L}_{\text{sink}}(\rho) + \mathcal{L}_{\text{deph}}(\rho), \tag{3}$$

where we introduced different decoherence processes

$$\mathcal{L}(\rho) = \sum_j \left[ L_j \rho L_j^\dagger - \frac{1}{2} \left\{ L_j^\dagger L_j, \rho \right\} \right] \tag{4}$$

with Lindblad operators $L_j$ as follows: Firstly

$$L_j^{\text{diss}} = \sqrt{\mu} \, \sigma_j^- \tag{5}$$

models small background losses, described here as independent spontaneous decay of all spins from the excited to the ground state. Note that the sum includes the target site $j = s$ ("the sink"). These loss processes compete with the transport of an initial excitation at a random initial site to the target site, where it is subsequently extracted. Even if the background losses are small they need to be taken into account as they set a natural upper bound $L_{\text{abs}}$ for the linear system dimension above which the transport from a random initial site will become highly inefficient, irrespective of dephasing. If the transport is dominated by nearest-neighbor hopping $J$, this length scale is given by

$$L_{\text{abs}} = \frac{J}{\mu} \tag{6}$$

Secondly, the extraction of excitations at the target site is modeled by an additional decay at that site with rate $\Gamma_s$.

$$L_j^{\text{sink}} = \sqrt{\Gamma_s} \, \sigma_s^-. \tag{7}$$

$\Gamma_s$ should be much larger than $\mu$, in order for excitations to be extracted rather than lost in the transport process. Yet it should not substantially exceed the nearest-neighbor hopping rate $J$ as this would suppress the final transfer to the target site. Thus we expect optimum efficiency conditions for

$$L \leq L_{\text{abs}}, \qquad \text{and} \qquad \mu \ll \Gamma_s \leq \mathcal{O}(J). \tag{8}$$

As we will show in the paper, the condition for the system size $L$ is in fact more stringent:

$$L \leq \sqrt{\frac{\Gamma_s}{\mu}} = L_{\text{abs}} \sqrt{\frac{\Gamma_s \mu}{J^2}}. \tag{9}$$

As an optimum value for $\Gamma_s$ is on the order of the hopping $J$ and the background absorption should be much slower than the hopping, the last term in the square root is less than unity and $L \leq \sqrt{J/\mu}$. Finally we assume a dephasing of the two-level transition at each site

$$L_j^{\text{deph}} = \sqrt{\gamma}\,\sigma_j^z. \tag{10}$$

We assume that the single initial excitation is created randomly and locally at any position (including the sink site) and all transport characteristics investigated below are understood as averaged over initial excitation positions.

The *efficiency* of the excitation transport can be expressed in terms of the population of the sink site, integrated over time:

$$\eta = \Gamma_s \int_0^\infty dt \, \langle \uparrow_s | \rho(t) | \uparrow_s \rangle = \Gamma_s \int_0^\infty dt \, \frac{1 + \langle \sigma_s^z(t) \rangle}{2}. \tag{11}$$

It describes the part of the initial excitation which is extracted from the sink at $t \to \infty$, while the rest is dissipated through the decay channels at the intermediate sites. In the absence of spontaneous decay an initial excitation can only leave through the the sink side and thus the transfer efficiency would be unity unless there are eigenstates of the Hamiltonian $H$ that are decoupled from the target site.

Another reasonable figure of merit is the *average transport time* [15], defined by:

$$\tau = \frac{\Gamma_s}{\eta} \int_0^\infty dt \, t \, \langle \uparrow_s | \rho(t) | \uparrow_s \rangle \tag{12}$$

Naively, one may not expect high efficiency if the transport time is large in the presence of a uniform steady dissipation, yet small times can not guarantee optimality.

## 3   Numerical methods

We first discuss the numerical simulation methods used in this work. Analyzing the dynamics of the transport process requires to solve the density-matrix equation of $L^2$ coupled spins. Although we restrict the analysis to a maximum of a single excitation, the numeric effort quickly grows with $L$. Thus we resort to the Monte-Carlo wave function (MCWF) approach introduced in [20], which trades precision for a smaller size of the computational problem. Secondly we introduce a Greens function (GF) description of the Lindblad equation, which is conveniently done using a vectorization of the single-particle density matrix, and gives direct access to time-integrated quantities such as transfer efficiency and transfer time, eqs.(11),(12), without having to solve the dynamical equations. It requires the diagonalization of an effective non-Hermitian Hamiltonian in an extended Hilbert space, which due to the restriction to single excitations can be performed numerically for intermediate system sizes. It furthermore forms the basis of analytical approximations.

### 3.1 Monte-Carlo wavefunction approach

For the numerical simulation of the dissipative dynamics of Eq. 3 we use the Monte-Carlo wave function method introduced in Ref. [20]. The algorithm is executed as follows. First of all, we sample an initial state as a spin-up state at a randomly chosen position. After that, the computational routine comprises two recursive steps:

1. Propagate the state vector for a small time $\delta t$ according to he non-Hermitian Hamiltonian:

$$H_{\text{nh}} = H - \frac{i}{2} \sum_i L_i^\dagger L_i \tag{13}$$

For a sufficiently small time increment $\delta t$ this yields:

$$|\tilde{\Psi}(t + \delta t)\rangle = (1 - i H_{\text{nh}} \delta t) |\Psi(t)\rangle$$

2. Generate a random number $x \in \{0, 1\}$ from a uniform distribution and compare it to:

$$p = \delta t \sum_i \langle \Psi(t) | L_i^\dagger L_i | \Psi(t)\rangle = \sum_i p_i.$$

If $x < p$ then one of the quantum jump operator $\{L_i\}$ is applied to the evolved wave function randomly with probability $p_i / p$ :

$$|\Psi(t + \delta t)\rangle = \frac{L_i |\tilde{\Psi}(t)\rangle}{||L_i |\tilde{\Psi}(t)\rangle||} \tag{14}$$

Otherwise, the wave function is simply normalized:

$$|\Psi(t + \delta t)\rangle = \frac{|\tilde{\Psi}(t)\rangle}{\sqrt{1 - p}}. \tag{15}$$

The algorithm assumes that the time step is sufficiently small to approach the exact solution. The aforementioned routine is performed until the finial time $T_{\text{fin}}$ is reached, providing a single quantum trajectory. Finally, the density matrix is reconstructed by averaging over a large number $N$ of realizations:

$$\rho(t) = \frac{1}{N} \sum_{k=1}^{N} |\Psi^k(t)\rangle \langle \Psi^k(t)|.$$

With the obtained density matrix the observables of interest are calculated. Note that although the upper time limit in the formal definition of the transfer efficiency (11) and time (12) is infinite, in practice a finite upper time $T_{\text{fin}} \approx 5/\mu$ is sufficient.

### 3.2 Single-particle Green's function approach

As an alternative to the MCWF method, we use an approach based on Greens function in an extended Hilbert space.

The master equation describing the time evolution of the single-particle density matrix $p$

$$p_{n,m}(t) \equiv \text{Tr} \left\{ \rho(t) \sigma_m^\dagger \sigma_n \right\} \tag{16}$$

reads ($\hbar = 1$):

$$
\begin{aligned}
\frac{\partial p_{n,m}(t)}{\partial t} \;=\;& -i\,[\mathbf{J},\mathbf{p}]_{nm} - \gamma\,(1-\delta_{nm})\,p_{nm} \\
& -\frac{\Gamma_s}{2}\Big(|s\rangle\langle s|\mathbf{p} + \mathbf{p}|s\rangle\langle s|\Big)_{nm} - \mu p_{nm},
\end{aligned}
\tag{17}
$$

where the matrix $\mathbf{J}$ is given by eq. (2). Here we introduced the projector $|s\rangle\langle s| \equiv |\!\uparrow_s\rangle\langle\uparrow_s|$ to the excited state of the target spin. Rearrangement of the dephasing term yields:

$$
\frac{\partial \mathbf{p}(t)}{\partial t} = -i\,[\mathbf{J},\mathbf{p}] - \gamma\left(\mathbf{p} - \sum_{k=1}^{L^2}|k\rangle\langle k|\mathbf{p}|k\rangle\langle k|\right) - \frac{\Gamma_s}{2}\Big(|s\rangle\langle s|\mathbf{p} + \mathbf{p}|s\rangle\langle s|\Big) - \mu\mathbf{p},
\tag{18}
$$

where $|k\rangle\langle k| \equiv |\!\uparrow_k\rangle\langle\uparrow_k|$ is the projector to the excited state of the $k$th spin.

We now use the notation $|\underline{\rho}\rangle$ as a vectorized form of an $m \times n$ matrix $\rho$:

$$
\begin{aligned}
|\underline{\rho}\rangle \;=\;& \Big[\rho_{11},\ldots,\rho_{m1},\rho_{12},\ldots,\rho_{m2},\ldots\rho_{1n},\ldots,\rho_{mn}\Big]^{\top}, \\
\langle\underline{\rho}| \;=\;& \Big[\rho_{11},\ldots,\rho_{m1},\rho_{12},\ldots,\rho_{m2},\ldots\rho_{1n},\ldots,\rho_{mn}\Big].
\end{aligned}
\tag{19}
$$

Following [21], the transformation rule for vectorized matrices reads $|U\cdot\rho\cdot V\rangle = \big(V^T \otimes U\big)|\underline{\rho}\rangle$, which we use to swap the order of operators, such that they act on the vectorized single-particle density matrix from the left in Eq. (18). This then results in a Schrödinger-type equation for the vectorized single-particle density matrix

$$
\frac{\partial}{\partial t}|\underline{\mathbf{p}}\rangle = -i\mathcal{H}\,|\underline{\mathbf{p}}\rangle,
\tag{20}
$$

where

$$
\mathcal{H} = \mathcal{H}_0 - iG
\tag{21}
$$

is a non-Hermitian Hamiltonian matrix in the doubled single-particle Hilbert space.

The Hermitian matrices $\mathcal{H}_0$ and $G$ are given by:

$$
\mathcal{H}_0 = \mathbf{1} \otimes \mathbf{J} - \mathbf{J} \otimes \mathbf{1}.
\tag{22}
$$

and

$$
G = \gamma\,(\mathbf{1} - \Pi) + \mu\mathbf{1} + \frac{\Gamma_s}{2}\Big(\mathbf{1} \otimes |s\rangle\langle s| + |s\rangle\langle s| \otimes \mathbf{1}\Big),
\tag{23}
$$

and describe the coherent transport and incoherent processes, respectively. Here we have introduced the projector $\Pi = \sum_{k=1}^{L^2} |k\rangle\langle k| \otimes |k\rangle\langle k|$. A formal solution of the equation of motion (20) reads $|\underline{\rho}(t)\rangle = e^{-i\mathcal{H}t}|\underline{\rho}(0)\rangle$. Inserting this in the definition of efficiency and transport time and using $\mathrm{Tr}(A^{\dagger}B) = \langle\underline{A}|\underline{B}\rangle$ one obtains an explicit expressions for the transfer efficiency

$$
\eta = \Gamma_s \int dt\,\langle\uparrow_s|\rho(t)|\!\uparrow_s\rangle = \Gamma_s \int dt\,\mathrm{Tr}(|\!\uparrow_s\rangle\langle\uparrow_s|\cdot\rho(t)) = \Gamma_s \int dt\,\langle\underline{\pi_s}|e^{-i\mathcal{H}t}|\underline{\rho}(0)\rangle =
$$
$$
= -i\Gamma_s\,\langle\underline{\pi_s}|\mathcal{H}^{-1}|\underline{\rho}(0)\rangle.
\tag{24}
$$

Here we have introduced the vectorized form of the projector $|s\rangle\langle s|$ to the target site: $|\underline{\pi_s}\rangle$. Likewise, for the transfer time:

$$
\tau = \frac{\Gamma_s}{\eta}\,\langle\underline{\pi_s}|\mathcal{H}^{-2}|\underline{\rho}(0)\rangle.
\tag{25}
$$

Note that the transfer time, $\tau$, can also be expressed as:

$$\tau = -\frac{1}{\eta}\frac{\partial \eta}{\partial \mu}.$$ (26)

Compared to the MCWF approach, the Green's function method has the striking advantage of not requiring the solution of a dynamic equation and providing solutions for any initial state once the inverted matrix $\mathcal{H}$ is evaluated. We noticed that the MCWF simulations require more computational efforts if $\gamma$ is increased, since the time step in the simulation necessarily decreases, while this is not the case for the Green's function method as it does not require to evaluate the dynamics. The drawback of the Green function method is, however, that the size of the matrix $\mathcal{H}^{-1}$ grows rapidly with the system size.

Finally, we use eq. (18) to derive a useful connection between the total background losses and the efficiency. After taking the trace on both sides of eq. (18):

$$\Gamma_s \int_0^\infty dt\, p_{s,s}(t) + \mu \int_0^\infty dt\, \mathrm{Tr}\big(\mathbf{p}(t)\big) = \eta + \mu \int_0^\infty dt\, \mathrm{Tr}\big(\mathbf{p}(t)\big) = 1.$$ (27)

This relation simply states a conservation law, namely that the excitation is either lost through homogeneous decay or is extracted at the target site. It assumes that at $t \to \infty$ the entire initial excitation has left the system, which in general requires $\mu > 0$. Eq. (27) can only be applied to the case $\mu = 0$ if there is no trapping in dark states in which case one finds $\eta = 1$. The efficiency is then no longer a good measure to characterize different transport scenarios. For a small background absorption rate $\mu$, the efficiency of the transport $\eta$ can be approximated by expanding $\eta(\mu)$ to lowest order in $\mu$ and using eq.(26)

$$\eta \approx 1 - \tau \cdot \mu.$$ (28)

This implies that a small transfer time $\tau$ guarantees the optimality of $\eta$ in the small-$\mu$ limit.

In the following sections, we examine two extreme cases: weak and strong incoherent processes. We will show that, in the case of weak dephasing and system sizes below a critical value, the transfer efficiency increases, whereas strong dephasing leads to a decrease in transfer efficiency. To support this, we first establish an upper bound on the transfer efficiency, which becomes exact in cases of weak or strong dephasing.

## 4 Transfer efficiency and transfer time: numerical results

Let us first present typical results of our numerical simulations for the transfer efficiency $\eta$. Fig. 2 shows $\eta$ obtained both from MCWF and GF simulations for different values of $\Gamma_s$ and for weak background absorption $\mu/J = 0.01$ as a function of dephasing $\gamma$. Note that the system size $L = 7$ is much below the absorption length $L_{\mathrm{abs}}$. One recognizes that there is an optimum value of the extraction rate $\Gamma_s/J = \mathcal{O}(1)$, as expected from eq. (8). One also notices, however, a non-monotonous dependence of $\eta$ on the dephasing. This non-monotonous behavior is characteristic for ENAQT. While the decrease of $\eta$ in the large dephasing limit is intuitive, the initial increase is not and will be explained in the following first. To this end we start by discussing the special case of $\mu = 0$.

### 4.1 Lattice without background losses ($\mu = 0$) and dark space

In order to understand the small transfer efficiency for values of $\gamma$ approaching zero, let us first set $\gamma = 0$ and look at the non-Hermitian Hamiltonian in the single-excitation Hilbert space,

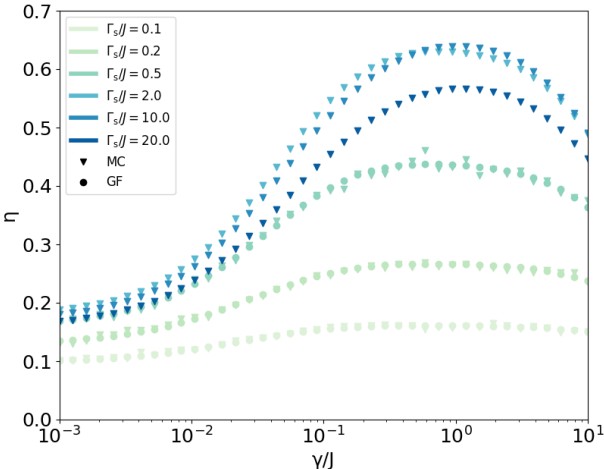

Figure 2: Efficiency as function of dephasing $\gamma$ for different extraction rates $\Gamma_s$ obtained from Monte-Carlo (MC) simulations and Green's function (GF) approach. The system has $L = 7$ sites in every direction and we have assumed open boundary conditions, with the sink site being at the center. $\mu/J = 0.01$ and thus $L_{abs} = 100 \gg L$.

which includes only hopping and the capturing of excitations at the sink:

$$H_{nh} = H - i\Gamma_s \left| s \right\rangle \left\langle s \right|. \tag{29}$$

For simplicity, we choose $\Gamma_s/J = 1$. In Fig. 3 the number of eigenstates of $H_{nh}$ are plotted ordered according to the imaginary part of the corresponding eigenvalue for a finite system of size $101 \times 101$.

One recognizes a very large fraction of states with vanishing imaginary part of the eigenvalue, which implies that they do not "drain" to the sink. This notion of "dark states" allows us to infer the upper limit for efficiency with dephasing "off". Nearly **90%** of all states are decoupled from the sink, which means that $\eta \leq 0.1$ for this extreme case. One can obtain a simple estimate for the number of dark states which explains their large number. For periodic boundary conditions, i.e. on a 2D torus, all eigenfunctions which have a node either in $x$ or in $y$ direction at the sink are dark modes. Due to the imaginary contribution to $H_{nh}$ at the sink site, only eigenstates which are symmetric superposition of Bloch waves around the sink in both directions have a non-vanishing imaginary part in the thermodynamic limit $L \to \infty$. This amounts to only **1/4** of all states. The remaining fraction of **3/4** are dark states. It can also be seen from the figure that the Hamiltonian has no damping gap between non-decaying and decaying states, so noise will destroy the uncoupled subspace for any $\gamma \neq 0$ with no threshold. This gives a qualitative explanation for the increase in efficiency by adding a small amount of dephasing, which is in line with arguments made for the case of an all-to-all coupling in Ref. [13].

Considering the arguments given above, for $\mu = 0$ the efficiency $\eta$ should approach unity in presence of any non-zero noise, rendering it a bad quantitative measure of transport. Instead, we may focus on a time benchmark of the system as plotted in Fig. 4. One can see that the transfer time drops drastically as the dephasing grows until $\gamma/J \approx 0.1$, then levels off in the region $0.1 < \gamma/J < 5$ and increases again for larger noise rates. For small dephasing the total transfer time is limited by the slow coupling of excitation from the dark-state manifold to states connected to the sink. As soon as the dephasing exceeds the nearest-neighbor hopping rate $J$, coherent transport starts to be suppressed, which eventually turns into incoherent hopping

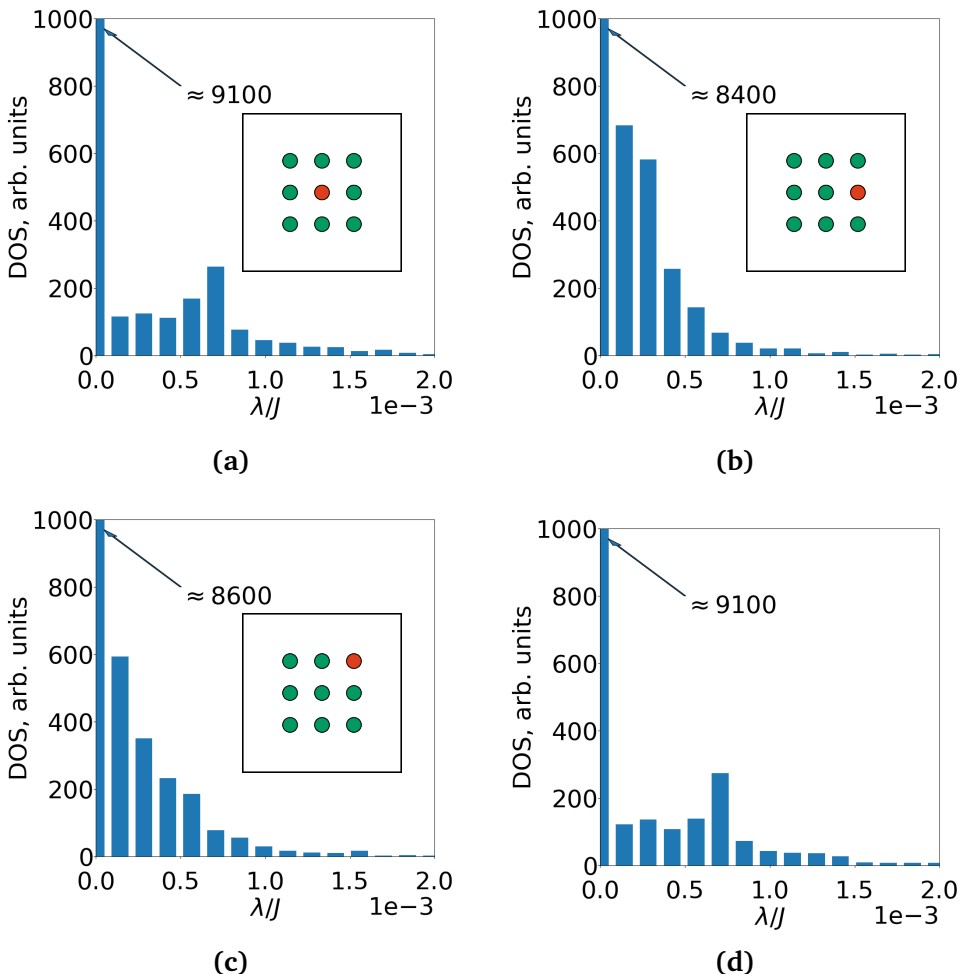

Figure 3: Density of eigenstates of $H_{\mathrm{nh}}$ ordered according to imaginary part of the corresponding eigenvalue for 101 × 101 lattice and the sink being (a) in the center, (b) displaced by one lattice constant in $x$, (c) displaced by one lattice constant along both $x$ and $y$, all with open boundary conditions. (d) Sink in the center but with periodic boundary conditions. There are 10201 eigenstates in total and $\Gamma_s/J = 1$.

with rate $\sim J^2/\gamma \ll J$. Note that the shortest possible transfer time is approximately $L/J$ which would correspond to $J\tau = 7$ in the figure.

As is the case for the efficiency in Fig. 2 the transfer time decreases with increasing $\Gamma_s$ until $\Gamma_s/J = \mathcal{O}(1)$. Beyond this value the last hopping process to the sink site becomes the bottleneck of the whole transfer process and the transfer time starts to increase again.

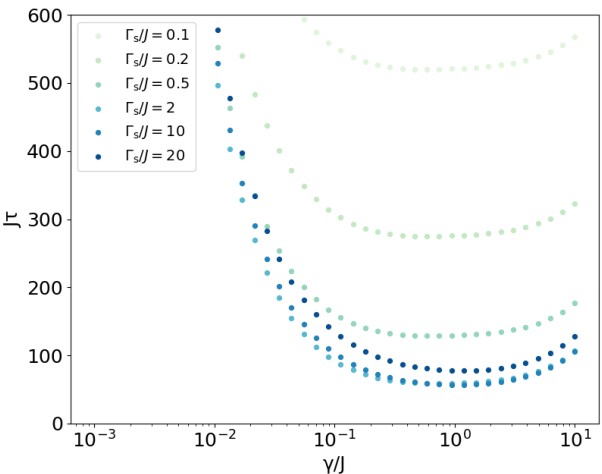

Figure 4: Dependence of transfer time of excitation on the strength of dephasing noise for a $7 \times 7$ lattice and $\mu = 0$.

## 4.2 General case, $\mu \neq 0$

Now we consider the case with both $\gamma$ and $\mu$ being nonzero. Fig. 5 demonstrates that transport efficiency drops as the rate of spontaneous decay grows. One can identify two regimes. For large values of the background decay rate $\mu$, such that $L_{\mathrm{abs}} = J/\mu < L$ there is a rapid power-law decrease of efficiency, almost independent on dephasing. Here a small to moderate dephasing has little to no effect on the transport efficiency. For small background decay a very small dephasing is sufficient to destroy the trapping of excitations in dark states and dephasing can lead to a sizeable increase of efficiency up to a system-size dependent maximum value (solid line) set by the background decay, which will be derived later on.

Finally let us discuss the behavior of the transfer time for finite backround absorption. In Fig. 6b we plot the dependence of $\tau$ on the dephasing rate $\gamma$ for different values of $\Gamma_s$ for a finite value of $\mu/J = 0.01$. For reference we also plot in Fig. 6a the efficiency for the same parameter from Fig. 2. One recognizes that in contrast to the loss-less case the transfer time first increases with $\gamma$ and attains a local maximum at about $\gamma/\mu \approx \mathcal{O}(1)$, followed by a minimum at $\gamma \approx J$. The minimum has a similar explanation as in the case without background losses. A non-vanishing value of $\gamma$ causes a coupling of the dark states to states with finite overlap with the sink site. So for $\gamma \lesssim \mu$ the coupling to the dark states is only little affected by dephasing, while the coherent transport between sites is slowed down. This explains the initial increase of $\tau$ with dephasing. However, when $\gamma$ exceeds $\mu$ it starts to increase the coupling rate to dark states and the transfer time becomes shorter again until the dephasing essentially kills coherent transport and turns it into an incoherent hopping with rate $\sim J^2/\gamma$.

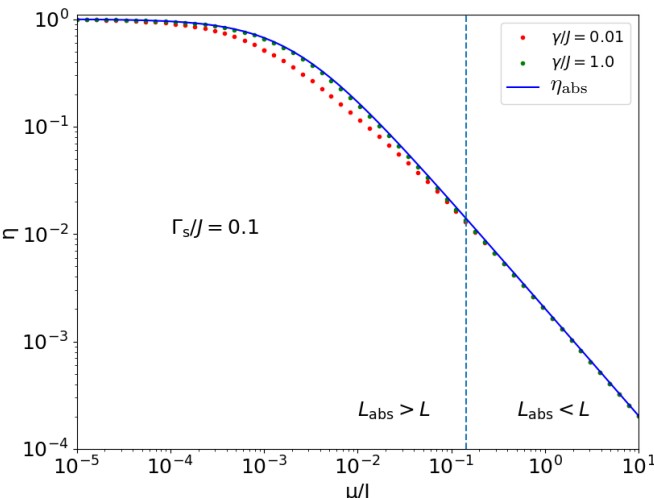

Figure 5: Reduction of transport efficiency with increasing decay rate. One notices that dephasing can only lead to an enhanced efficiency if the background loss is sufficiently small such that $L_{abs} > L$. The vertical dashed line indicates where $L_{abs} = L$. Also shown is a system-size dependent upper bound to the efficiency due to background losses, $\eta_{abs}$, derived later. Here $L = 7$.

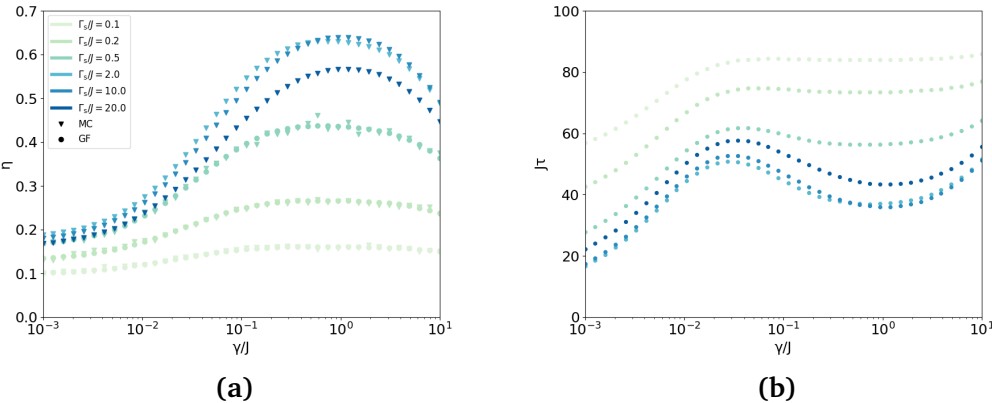

Figure 6: (a) Efficiency $\eta$ as function of dephasing from Fig. 2. (b) Transfer time $\tau$ for non-vanishing background decay $\mu/J = 0.01$ as function of dephasing obtained from Green's function (GF) method. The size of the system is $7 \times 7$ sites.

## 5   Analytic approach

In the following we derive analytic bounds for the efficiency in a 2D lattice. For a 1D system, compact explicit expressions can be obtained, which are given in the Appendix. As was mentioned above, we consider the case when the initial state $|\underline{\rho}\,(0)\rangle$ has a single excitation assumed to be uniformly distributed across all lattice sites. In this important case, a concise expression for the transport efficiency can be derived using

$$\mathcal{H}|\underline{\rho}\,(0)\rangle = \left(\mathcal{H}_0 - iG\right)|\underline{\rho}\,(0)\rangle = -iG\,|\underline{\rho}\,(0)\rangle = -i\mu|\underline{\rho}\,(0)\rangle - i\frac{\Gamma_s}{L^2}|\underline{\pi}_s\rangle.$$

As shown in detail in the Appendix, one finds:

$$\eta = 1 - \mu L^2 \left\langle \underline{\rho}(0) \left| \frac{1}{G + \mathcal{H}_0 G^{-1} \mathcal{H}_0} \right| \underline{\rho}(0) \right\rangle, \tag{30}$$

Notably the transport efficiency can also be rewritten as:

$$\eta = \frac{\Gamma_s}{\mu L^2} \left( 1 - \Gamma_s \left\langle \underline{\pi_s} \left| \left[ \frac{1}{G + \mathcal{H}_0 G^{-1} \mathcal{H}_0} \right] \right| \underline{\pi_s} \right\rangle \right). \tag{31}$$

Both forms will be used in the following.

## 5.1 Limitation on system size from background absorption

In the introduction we have argued that there is an upper limit for the system size in the presence of a small, but non-vanishing background absorption rate $\mu$ up to which efficient transfer can be expected in a 2D lattice with short-range hopping. Since the initial excitation can take place at any lattice site with equal probability the average propagation distance to the sink, which is of order of the linear lattice dimension $L$ should be less than the naive absorption length $L_{\text{abs}} = J/\mu$. We will now demonstrate that there is in fact a more stringent limitation on system size that can be obtained by considering a simple yet non-trivial upper bound for the efficiency $\eta$, following directly from eq.(30). Indeed, by using that

$$H_{\text{eff}} = G + \mathcal{H}_0 G^{-1} \mathcal{H}_0 \tag{32}$$

is a positive definite matrix and employing the Cauchy-Schwarz inequality, one finds

$$\frac{1}{L^4} = \langle \underline{\rho}(0) | \underline{\rho}(0) \rangle^2 = \langle \underline{\rho}(0) | H_{\text{eff}}^{-1/2} H_{\text{eff}}^{1/2} | \underline{\rho}(0) \rangle^2 \le \langle \underline{\rho}(0) | H_{\text{eff}}^{-1} | \underline{\rho}(0) \rangle \langle \underline{\rho}(0) | H_{\text{eff}} | \underline{\rho}(0) \rangle,$$

where in the first equality we have used that the initial excitation is equally distributed. This then yields

$$\eta \le 1 - \mu \frac{1}{L^2 \langle \underline{\rho}(0) | H_{\text{eff}} | \underline{\rho}(0) \rangle} = \frac{\Gamma_s}{L^2 \mu + \Gamma_s} \equiv \eta_{\text{abs}}. \tag{33}$$

The right hand side is an on first glance surprising upper bound and can be interpreted as follows: It corresponds to the efficiency for the case where an excitation becomes rapidly distributed evenly over all sites and stays evenly distributed for all times. In this case the (uniform) excitation density decays as $n_j(t) = n(t) = \exp(-t\Gamma_{\text{tot}})$, where $\Gamma_{\text{tot}} = \Gamma_s + \mu L^2$ is the total decay rate of the system. Then, according to eq. (27) one has $\eta = 1 - \mu \int_0^\infty dt \, \text{Tr}\underline{\rho}(t) = \frac{\Gamma_s}{\Gamma_s + \mu L^2}$. It shows that the efficiency drops with the system size as $\frac{1}{L^2}$ for large $L$ and that efficient extraction of excitation requires linear system sizes small compared to

$$L \le \sqrt{\frac{\Gamma_s}{\mu}} = L_{\text{abs}} \sqrt{\frac{\Gamma_s \mu}{J^2}} \tag{34}$$

Under optimum conditions $\Gamma_s \sim \mathcal{O}(J)$ and $\mu \ll J$ and then the right hand side is much smaller than $L_{\text{abs}}$.

Making use of the relation between transfer efficiency and time, eq.(26), one can also establish a lower bound to the transfer time due to back ground absorption

$$\tau \ge \frac{L^2}{L^2 \mu + \Gamma_s}. \tag{35}$$

Contrary to the efficiency the lower limit for the transfer time becomes size independent for large values of $L$. The limits (33) and (35) can be understood as follows: As illustrated in Fig. 7 all initial excitations further away from the extraction center than $\sqrt{\Gamma_s/\mu}$ will be lost by background absorption and do not contribute to the extraction process, resulting in an asymptotic $1/L^2$ scaling of $\eta$. Only excitations inside the circle reach the sink and the time for this to take place becomes independent on system size. An important conclusion to light-harvesting systems is that the optimum size of an "antenna" connected to a single extraction site is $L_{\text{opt}} \sim \sqrt{\Gamma_s/\mu}$.

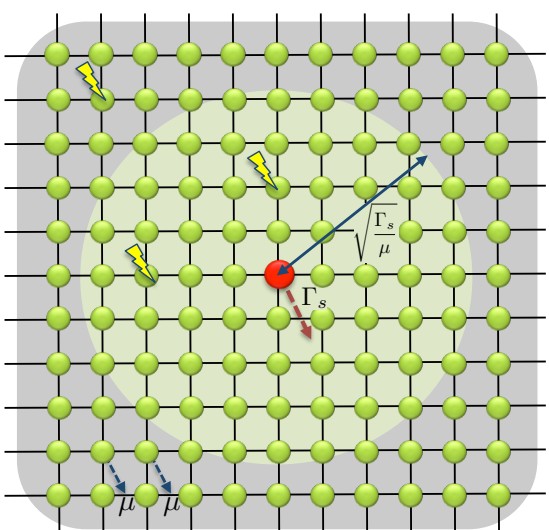

Figure 7: Illustration of effective spatial range of excitation capture at sink site (red). Initial excitations further away from extraction center than $\sqrt{\Gamma_s/\mu}$ will effectively be lost by background absorption and do not contribute.

From the derivation of eq. (33) we expect that $\eta_{\text{abs}}$ becomes actually an accurate estimate for the efficiency for small $\Gamma_s$. Indeed, for small $\Gamma_s$ the initial state $|\rho(0)\rangle$ is an eigenstate of $H_{\text{eff}}$ and therefore the Cauchy-Schwarz inequality becomes an equality. However, in this limit the efficiency is small. In Fig. 5 we compare numerical results with this upper bound. One notices that the upper bound becomes an excellent approximation to $\eta$ in the low-efficiency regime $L \gg L_{\text{abs}} > (\Gamma_s/\mu)^{1/2}$. However, in the high-efficiency regime, $L < L_{\text{abs}}$, the actual value of $\eta$ is significantly below the bound. Most importantly it can be *increased* by adding a finite dephasing $\gamma$. This is the regime of interest for us and will be discussed in detail in the following.

## 5.2 Upper bounds to transfer efficiency

In the following we will present a quantitative analytic approach to the transport efficiency in the two limits of small and large dephasing. For small dephasing, coherence effects are dominant and we will see that here dephasing leads to an increase of the efficiency of the transport process since is destroys destructive interferences. The limit of large dephasing on the other hand can be understood as a Zeno-type suppression of transport due to rapid projections on spin eigenstates.

First we assume that the unitary evolution of the system is only slightly perturbed by the losses and dephasing processes. We will show that, for large hopping amplitude, when the system starts its evolution from the maximally mixed state, i.e. all lattice sites are equally populated, the efficiency of the transport $\eta$ always initially increases with the dephasing rate

$\gamma$ in any system.

For the next step we assume that the dephasing rate $\gamma$ is much larger than other parameters in the system. In this limiting case the excitation becomes more and more localized at its initial site due to the Zeno effect, and the efficiency of transport $\eta$ always drops with $\gamma$.

The drawback of the limit (33) is that it does not describe the dependence on $\gamma$ and $J$. To address this point, we will derive in the following more precise estimates of the transfer efficiency, which depend on these parameters. We derive different estimates in the two limits of weak and strong dephasing that describe the behavior of $\eta$ rather accurately and explain quantitatively the existence of a regime of environment assisted, increased transfer efficiency.

### 5.2.1 Weak-dephasing, "coherent" regime

We start by deriving an upper bound to the efficiency which will turn out to be a rather accurate approximation in the regime of small dephasing. To begin with we note that equation (30) can be rewritten as

$$\eta = 1 - \mu L^2 \left\langle \underline{\rho}\,(0) \left| G^{-1/2} \frac{1}{1 + \left(G^{-1/2}\mathcal{H}_0 G^{-1/2}\right)^2} G^{-1/2} \right| \underline{\rho}\,(0) \right\rangle, \qquad (36)$$

where we have used the fact that $G$ is a positive definite matrix. The most important contribution in the weak dephasing limit comes from states in the null space $\{|\underline{\Phi}_\alpha\rangle\}$ of the matrix $G^{-1/2}\mathcal{H}_0 G^{-1/2}$. defined by

$$\mathcal{H}_0 G^{-1/2}|\underline{\Phi}_\alpha\rangle = 0.$$

Assuming first that the hopping matrix $J$ has a non-degenerate spectrum, the vectors $|\underline{\Phi}_\alpha\rangle$ can be expressed as

$$|\underline{\Phi}_\alpha\rangle = G^{1/2}|\alpha\rangle \otimes |\alpha\rangle,$$

with $|\alpha\rangle$ being eigenvectors of $J$. In the case of degeneracy of $J$, additional states would appear in the null space of $G^{-1/2}\mathcal{H}_0 G^{-1/2}$. However, these additional states do not affect the upper bounds for the efficiency, but may affect the quality of the bound in the sense that they slightly enhance the difference between true values and the bound. We note that the states $|\underline{\Phi}_\alpha\rangle$ are in general not orthonormal. After applying the Gram-Schmidt orthogonalisation procedure a new set $\{|\underline{\Psi}_\alpha\rangle\}$ is obtained, which again forms the null space of the matrix $G^{-1/2}\mathcal{H}_0 G^{-1/2}$

$$|\underline{\Psi}_\alpha\rangle = \sum_\nu \Omega_{\nu\alpha}^{-1/2} G^{1/2}\big(|\alpha\rangle \otimes |\alpha\rangle\big),$$

where $\Omega_{\alpha\nu}^{-1/2}$ are the matrix elements of the inverse square root of

$$\Omega_{\alpha\beta} = \big(\langle\alpha| \otimes \langle\alpha|\big)G\big(|\beta\rangle \otimes |\beta\rangle\big) = \big(\mu + \Gamma_s\,|\langle\alpha|\,s\rangle|^2\big)\delta_{\alpha\beta} + \gamma\big(\delta_{\alpha\beta} - W_{\alpha\beta}\big), \qquad (37)$$

where $\alpha, \beta = 1, \ldots, L^2$ and

$$W_{\alpha\beta} = \sum_k |\langle\alpha|\,k\rangle|^2\,|\langle\beta|\,k\rangle|^2.$$

We note that $W_{\alpha\beta} \geq 0$, and $\sum_\alpha W_{\alpha\beta} = \sum_\beta W_{\alpha\beta} = 1$. That is, $W$ is a double stochastic matrix.

It is well known that the maximum eigenvalue of a doubly stochastic matrix is equal to 1 [22] In other words, for such a matrix, we have: $W \leq 1$.

Then, according to the spectral decomposition theorem:

$$\frac{1}{1 + \left(G^{-1/2}\mathcal{H}_0 G^{-1/2}\right)^2} > \sum_\beta |\underline{\Psi}_\beta\rangle\langle\underline{\Psi}_\beta|,$$

Combining these with eq. (32), we find that

$$
\begin{aligned}
H_{\text{eff}}^{-1} &\geq \sum_{\alpha,\beta,\nu} \left(\Omega^{-1/2}\right)_{\alpha\beta} \left(\Omega^{-1/2}\right)_{\nu\beta} (|\alpha\rangle \otimes |\alpha\rangle)(\langle\nu| \otimes \langle\nu|) \\
&= \sum_{\alpha,\nu} \left(\Omega^{-1}\right)_{\alpha\nu} (|\alpha\rangle \otimes |\alpha\rangle)(\langle\nu| \otimes \langle\nu|).
\end{aligned}
$$

Substituting this into the expression fo $\eta$ gives:

$$
\begin{aligned}
\eta &= 1 - \mu L^2 \langle \underline{\rho}(0)| \frac{1}{H_{\text{eff}}} |\underline{\rho}(0)\rangle \\
&\leq 1 - \mu L^2 \sum_{\alpha,\nu} \left(\Omega^{-1}\right)_{\alpha\nu} \langle \underline{\rho}(0)| \Big(|\alpha\rangle \otimes |\alpha\rangle\Big)\Big(\langle\nu| \otimes \langle\nu|\Big)|\underline{\rho}(0)\rangle \\
&= 1 - \frac{\mu}{L^2} \sum_{i,j} \sum_{\alpha,\nu} \left(\Omega^{-1}\right)_{\alpha\nu} \left|\langle i |\alpha\rangle\right|^2 \left|\langle j |\nu\rangle\right|^2
\end{aligned}
$$

This then yields an upper bound $\eta_{\text{coh}}$ for the fidelity in the coherent regime, sharper than eq.(33).

$$
\eta \leq \eta_{\text{coh}} = 1 - \frac{\mu}{L^2} \sum_{\alpha,\nu} \left(\Omega^{-1}\right)_{\alpha\nu}. \tag{38}
$$

This is the first main result of this section. We note that for small decoherence rates ($\mu, \Gamma_s, \gamma \ll J$), this upper bound becomes asymptotically exact. An important implication of Eq. (38) is that for small values of $\gamma$ the efficiency of the transfer process increases as the dephasing rate $\gamma$ grows. Using the relation for the derivative of a matrix $\frac{\partial}{\partial\gamma}\Omega^{-1} = -\Omega^{-1}\frac{\partial\Omega}{\partial\gamma}\Omega^{-1}$ and noting that because $W \leq 1$ the derivative of the matrix $\Omega$ with respect to $\gamma$ is positive definite we find that

$$
\frac{\partial \eta_{\text{coh}}}{\partial \gamma} \geq 0. \tag{39}
$$

This is the second main result of this section. It shows that under conditions of small decoherence, i.e. in the coherent regime, adding small dephasing leads to an increase of the fidelity!

Finally the upper bound, eq.(38), also provides a quantitative understanding of the limited efficiency in the absence of dephasing, i.e. for $\gamma = 0$ and small background losses, $\mu \to 0$. In this limit the inequality (38) becomes an equality:

$$
\eta = 1 - \frac{1}{L^2} \sum_{\alpha} \frac{\mu/\Gamma_s}{|\langle\alpha| s\rangle|^2 + \mu/\Gamma_s}. \tag{40}
$$

For $\mu \to 0$ only states $|\alpha\rangle$ contribute in the above sum which have zero overlap with the target site, i.e. for which $|\langle\alpha|s\rangle| = 0$, and the term in the sum is equal to one. Thus one finds

$$
\eta \to 1 - \frac{N_{\text{dark}}}{L^2}, \qquad \text{for} \quad \mu \to 0, \tag{41}
$$

where $N_{\text{dark}}$ is the number of eigenstates (dark states) of $J$ with vanishing overlap with the sink.

### 5.2.2 Large-noise or "Zeno" regime

The transport dynamics can be well understood if the dephasing rate $\gamma$ dominates over the nearest-neighbor hopping, $\gamma \gg J$. In this limit the transport process becomes fully incoherent and can be described by rate equations. The excitation transport then corresponds to a classical

random walk with background losses. The corresponding transport rate between adjacent sites is given by

$$J_{\text{inc}} \sim \frac{J^2}{\gamma + \mu} \ll J. \tag{42}$$

Thus the transport process is slowed down substantially as compared to the coherent case. The competition between transport and background absorption then leads to a much shorter absorption length in the strong-dephasing limit:

$$L_{\text{abs}}^{\text{incoh}} = \frac{J^2}{(\gamma + \mu)\mu} \ll L_{\text{abs}}. \tag{43}$$

As $J_{\text{inc}} \sim \gamma^{-1}$ the transfer efficiency will decrease with increasing dephasing, which gives an intuitive explanation of the numerical results in Fig. 2 in the regime $\gamma/J \gg 1$. We now want to derive an upper bound on $\eta$ in the large dephasing regime to get a quantitative understanding of the transport in this regime.

The quality of the estimate (38), derived in the previous subsection, deteriorates as the dephasing rate $\gamma$ increases, particularly when $\gamma$ approaches or exceeds the hopping $J$. For this reason we here derive an alternative upper bound, which describes the transfer process better at large $\gamma$. To this end we first note that the matrix $G + \mathcal{H}_0 G^{-1} \mathcal{H}_0$ can be bounded from above as follows:

$$G + \mathcal{H}_0 G^{-1} \mathcal{H}_0 \leq G + \mathcal{H}_0 G_0^{-1} \mathcal{H}_0,$$

where

$$G_0 = \gamma(1 - \Pi) + \mu \mathbb{1}$$

is the part of $G$ that only accounts for background losses and dephasing. Then following similar steps as in the previous sub-section and using the form (31) for $\eta$ we find

$$
\begin{aligned}
\eta &= \frac{\Gamma_s}{\mu L^2} \left( 1 - \Gamma_s \langle \underline{\pi_s} | \left[ \frac{1}{G + \mathcal{H}_0 G^{-1} \mathcal{H}_0} \right] |\underline{\pi_s}\rangle \right) \\
&\leq \frac{\Gamma_s}{\mu L^2} \left( 1 - \Gamma_s \langle \underline{\pi_s} | \left[ \frac{1}{\gamma(1-\Pi) + D} \right] |\underline{\pi_s}\rangle \right),
\end{aligned}
$$

where

$$D = \frac{\Gamma_s}{2} \left( \mathbb{1} \otimes |s\rangle\langle s| + |s\rangle\langle s| \otimes \mathbb{1} \right) + \mu \mathbb{1} + \mathcal{H}_0 G_0^{-1} \mathcal{H}_0$$

is a positive definite matrix. Then

$$\frac{1}{\gamma(1-\Pi) + D} = D^{-1/2} \left( \frac{1}{1 + \gamma D^{-1/2}(1-\Pi) D^{-1/2}} \right) D^{-1/2}.$$

Similarly to the discussion in the previous subsection we are looking for the null space $\{|\underline{\theta_k}\rangle\}$ of the matrix $D^{-1/2}(1-\Pi) D^{-1/2}$, which can be constructed by

$$|\underline{\theta_k}\rangle = D^{1/2} |k\rangle \otimes |k\rangle.$$

The vectors $|\underline{\theta_k}\rangle$ are non-orthonormal. After orthogonalization one obtains a new set $|\underline{\varphi_k}\rangle$ which forms the null space. Then according to the spectral decomposition theorem, we arrive at:

$$
\begin{aligned}
\frac{1}{1 + \gamma D^{-1/2}(1-\Pi) D^{-1/2}} &> \sum_k |\underline{\varphi_k}\rangle\langle\underline{\varphi_k}| \\
&= \sum_{n,k,m} \left( D^{-1/2} \right)_{kn} \left( D^{-1/2} \right)_{mn} D^{1/2} \Big( |k\rangle \otimes |k\rangle \Big) \Big( \langle m| \otimes \langle m| \Big) D^{1/2}.
\end{aligned}
$$

And therefore

$$\frac{1}{\gamma(1-\Pi)+D} \geq \sum_{nkm} \left(D^{-1/2}\right)_{kn} \left(D^{-1/2}\right)_{mn} \Big(|k\rangle \otimes |k\rangle\Big)\Big(\langle m| \otimes \langle m|\Big)$$

Taking the expectation value of this expression in the state $|\underline{\pi_s}\rangle$ then leads to the new upper bound for $\eta$

$$\eta \leq \eta_{\text{incoh}} = \frac{\Gamma_s}{\mu L^2}\Big(1-\Gamma_s\left(D^{-1}\right)_{ss}\Big), \tag{44}$$

The explicit form of the matrix elements of $D$ in the basis of the lattice coordinates $\{|\underline{m}\rangle = |m\rangle \otimes |m\rangle\}$ is given by:

$$D_{mn} = \Gamma_s \delta_{sm}\delta_{sn} + \mu\delta_{nm} + \langle\underline{m}|\mathcal{H}_0 G_0^{-1}\mathcal{H}_0|\underline{n}\rangle.$$

The bound (44) can be simplified further (see Appendix).

$$\eta_{\text{incoh}} = \frac{\Gamma_s}{\mu L^2}\frac{1}{1+\Gamma_s\langle s|\frac{1}{\mu+K}|s\rangle}, \tag{45}$$

where

$$K_{mn} = \frac{2}{\mu+\gamma}\Big(\delta_{nm}\langle m|J^2|n\rangle - \langle m|J|n\rangle^2\Big).$$

One can show, the right-side of the inequality (45) matches the expression one can obtain from a rate-equation approach. It decreases with increasing dephasing rate. Eq.(45) is the main result of this section. It gives a quantitative upper bound $\eta_{\text{incoh}}$ to the efficiency in the large-dephasing regime.

Finally we note that also from this bound one can derive the upper limit set by background absorption, eq.(33). To see this we note that the matrix $K$ is positive semidefinite with the $L^2$ dimensional null vector $|e\rangle = \frac{1}{L}(1,1,...1)^T$ and thus $\frac{1}{\mu+K} \geq \frac{1}{\mu}|e\rangle\langle e|$. Consequently

$$\eta_{\text{incoh}} = \frac{\Gamma_s}{\mu L^2}\frac{1}{1+\Gamma_s\langle s|\frac{1}{\mu+K}|s\rangle} \leq \frac{\Gamma_s}{\mu L^2}\frac{1}{1+\frac{\Gamma_s}{\mu}|\langle s|e\rangle|^2} = \frac{\Gamma_s}{\Gamma_s+\mu L^2} = \eta_{\text{abs}}.$$

## 5.3 Efficiency estimate and optimum dephasing

Combining eqs. (38) and (45) we arrive at the following estimate for the efficiently of the transport

$$\eta \leq \min\Big\{\eta_{\text{coh}}, \eta_{\text{incoh}}\Big\} = \min\left\{1-\frac{\mu}{L^2}\sum_{\alpha,\nu}\left(\Omega^{-1}\right)_{\alpha\nu} , \frac{\Gamma_s}{\mu L^2}\frac{1}{1+\Gamma_s\langle s|\frac{1}{\mu+K}|s\rangle}\right\}. \tag{46}$$

As noted above, this bound matches the exact efficiency of the transfer process for both small and large values of the dephasing rate $\gamma$ rather well. Fig. 8 shows the exact numerical value of $\eta$ alongside the estimate (46) as a function of $\gamma$ for the case of dipolar hopping $J_{ij} = \frac{J}{|i-j|^3}$ ($i \neq j$) and for following parameters: $L = 5, \Gamma_l/J = 0.1, \mu/J = 0.01$. As seen in Fig. 8, the estimate (46) closely matches the behavior of $\eta$ as a function of $\gamma$.

One notices that in general there is an optimum value $\gamma_{\text{opt}}$ of the dephasing rate for which the efficiency is maximal. Eq. (46) identifies the range of $\gamma$ values where there is an enhancement of quantum transport. The first argument on the right-hand side of (46) increases with $\gamma$, while the second argument decreases. Therefore, by setting these two expressions equal, one can calculate the crossing point $\gamma_0$ which is close to the optimum value $\gamma_{\text{opt}}$. In general

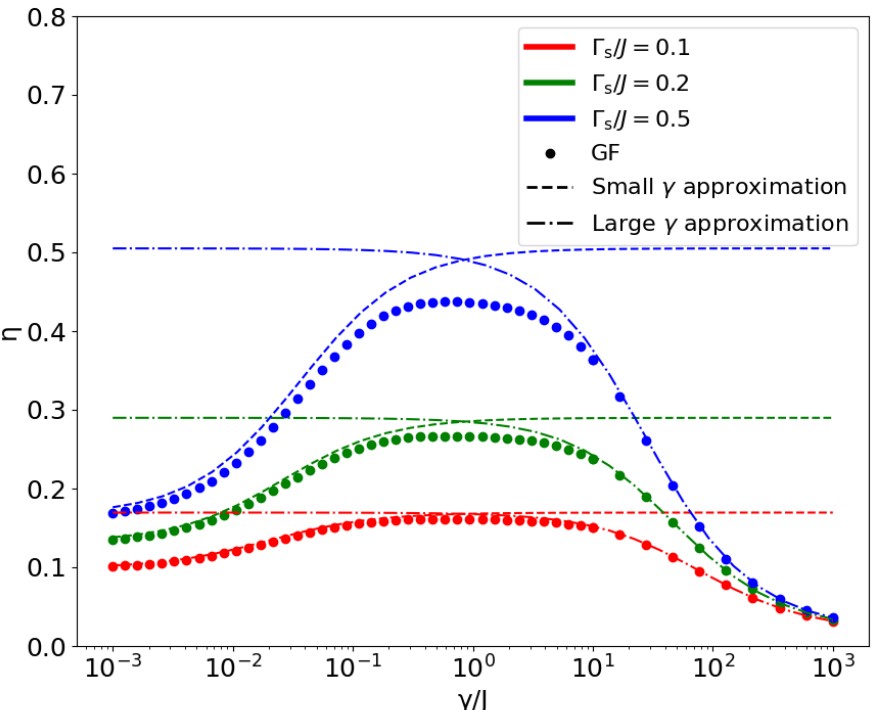

Figure 8: Comparison of efficiency as a function of dephasing for simulations (dots) and analytic approximations addressing cases of weak and strong dephasing (dash and dash-dotted lines respectively). $\mu/J = 0.01$, $L = 5$

one finds that $\gamma_0$ decreases with increasing system size $L$ eventually becoming zero at some critical value beyond which ENAQT ceases to exist. As shown in the Appendix one can derive an explicit expression for $\gamma_0$ for the 1D case, which is the solution of:

$$\frac{\Gamma_s \left(\gamma_0 L + (L+1)\mu\right)}{2\mu\left(\Gamma_s + \frac{L+1}{2}\mu\right) + \gamma_0\left(\Gamma_s + L\mu\right)} = \frac{\Gamma_s}{\mu L} \frac{1}{1 + \frac{\Gamma_s}{\mu L}\left(1 + \frac{\mu}{\mu + \frac{4J^2}{\mu+\gamma_0}}\frac{L^2-1}{6}\right)}. \tag{47}$$

The positive root of this equation for a large $L \gg 1$ and small $\mu \ll J$ takes on the simple form

$$\gamma_0 \approx \frac{5J}{L} - \mu. \tag{48}$$

In Fig. 9 we have plotted both $\gamma_0$ from eq.(48) and the numerical optimum value $\gamma_{\text{opt}}$ as a function of system size for a 1D chain.

In conclusion we see that the derived bounds for the transfer efficiency in the low and high dephasing limits give an accurate quantitative description of the effect of dephasing. In particlar they show that there is a substantial enhancement of quantum transport by small dephasing, i.e. ENAQT, but it is limited to system sizes below a length of order

$$L \leq \mathcal{O}(1)\frac{J}{\mu}. \tag{49}$$

Although the analytic estimates on the optimum dephasing for maximum transfer efficiency were derived from a one-dimensional model, we expect them to hold also in the 2D case, as the shortest path from a random initial site to the sink in a 2D lattice scales $\mathcal{O}(1)L$.

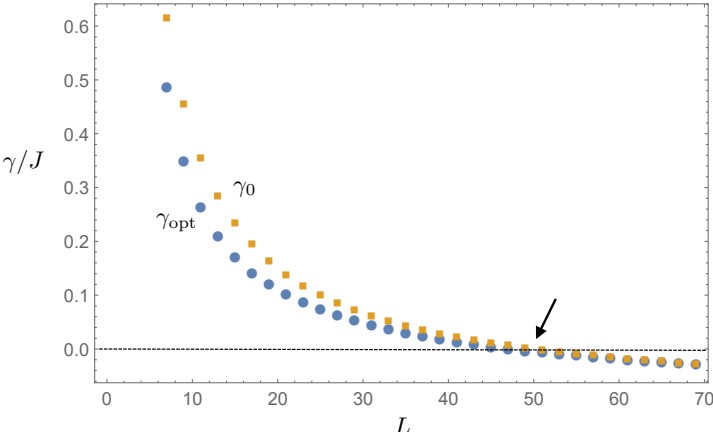

Figure 9: Optimal dephasing rate $\gamma_{\text{opt}}$ (blue circles) as well as analytic approximation eq. (47) (orange squares) as a function of the size $L$ for a 1D lattice. The arrow indicates the point at which $\gamma_0$ becomes negative. The parameters are as follows: $\Gamma/J = 0.1$, $\mu/J = 0.1$. We have chosen a relatively large $\mu$ to get a small critical length.

## 6   Conclusion

We studied the phenomenon of environment-assisted quantum transport (ENAQT) of an initial excitation in a two-dimensional lattice of size $L \times L$ and with weak background absorption. The initial excitation ocurs at a random site and we consider transport to a specific target site. We showed that a small amount of dephasing can enhance the transport efficiency if the system size is below some critical value. Under this condition we identified two regimes, characteristic of ENAQT: (i) a coherent regime of small-dephasing, where the efficiency increases with the dephasing rate, and (ii) a Zeno regime of large dephasing, where the transport becomes a classical random walk with a hopping rate inversely proportional to the dephasing strength and the efficiency decreases with dephasing. The initial increase of efficiency can be traced back to the existence of a large number of eigenstates of the transport Hamiltonian with vanishing overlap with the target site. Dephasing induces a coupling of this dark space to states with overlap to the target site. Most importantly we derived tight upper bounds for the transfer efficiency in the two regimes. These analytic expressions provide a quantitative description of ENAQT in 2D lattice systems. Furthermore they allow to identify several conditions for high transfer efficiency: (1) Due to background absorption, the transfer efficiency decreases as $1/L^2$ for system sizes $L > \sqrt{\Gamma_s/\mu}$, where $\mu$ is the uniform rate of background absorption and $\Gamma_s$ the rate of extraction at the target site. Thus excitation harvesting in a system with linear dimension larger than $\sqrt{\Gamma_s/\mu}$ is always ineffecient. (2) Dephasing leads to an enhancement of the transfer efficiency up to an optimum value $\gamma_{\text{opt}} \approx cJ/L - \mu$, where $c = \mathcal{O}(1)$ is a numerical factor of order one, by lifting the decoupling of dark states from the target site.

## Appendix

### A.   Derivation of expressions (30) and (31) for the efficiency $\eta$

We first note that the initial state $|\rho(0)\rangle$ with equal probability at every site is an eigenstate of the hopping Hamiltonian $\mathcal{H}_0 = \mathbb{1} \otimes J - J \otimes J$ with eigenvalue zero, i.e. it belongs to the null

space

$$\mathcal{H}_0 |\underline{\rho}(0)\rangle = 0.$$

Then one finds using the definition of $G$, eq.(23)

$$\mathcal{H}|\underline{\rho}(0)\rangle = (G - i\mathcal{H}_0)|\underline{\rho}(0)\rangle = \mu|\underline{\rho}(0)\rangle + \frac{\Gamma_s}{L^2}|\underline{\pi}_s\rangle.$$

In the last line we used $(1 - \Pi)|\underline{\rho}(0)\rangle = 0$ and $(1 \otimes |s\rangle\langle s| + |s\rangle\langle s| \otimes 1)|\underline{\rho}(0)\rangle = \frac{2}{L^2}|\underline{\pi}_s\rangle$. This then yields

$$
\begin{aligned}
\eta & = -i\Gamma_s \langle\underline{\pi}_s|\frac{1}{\mathcal{H}}|\underline{\rho}(0)\rangle = \Gamma_s \langle\underline{\pi}_s|\frac{1}{G + i\mathcal{H}_0}\Big|\underline{\rho}(0)\Big\rangle \\
& = 1 - L^2\mu\Big\langle\underline{\rho}(0)\Big|\frac{1}{G + i\mathcal{H}_0}\Big|\underline{\rho}(0)\Big\rangle.
\end{aligned}
$$

Using the facts that $\eta$ is real and $G$ is a positive-definite matrix, we obtain

$$
\begin{aligned}
\eta & = 1 - \frac{1}{2}L^2\mu\langle\underline{\rho}(0)|\left(\frac{1}{G + i\mathcal{H}_0} + \frac{1}{G - i\mathcal{H}_0}\right)|\underline{\rho}(0)\rangle \\
& = 1 - \frac{1}{2}L^2\mu\langle\underline{\rho}(0)|G^{-1/2}\left(\frac{1}{1 + iG^{-1/2}\mathcal{H}_0 G^{-1/2}} + \frac{1}{1 - iG^{-1/2}\mathcal{H}_0 G^{-1/2}}\right)G^{-1/2}|\underline{\rho}(0)\rangle \\
& = 1 - L^2\mu\langle\underline{\rho}(0)|G^{-1/2}\left(\frac{1}{1 + G^{-1/2}\mathcal{H}_0 G^{-1}\mathcal{H}_0 G^{-1/2}}\right)G^{-1/2}|\underline{\rho}(0)\rangle \\
& = 1 - L^2\mu\langle\underline{\rho}(0)|\frac{1}{G + \mathcal{H}_0 G^{-1}\mathcal{H}_0}|\underline{\rho}(0)\rangle,
\end{aligned}
$$

which coincides with Eq. (30) in the main text of the paper. In the same way, Eq. (31) can be derived.

## B. Derivation of the upper bound (45)

In Sect.5.2.2 we have shown that in the large-noise regime one can derive the following upper bound to the efficiency:

$$\eta \leq \frac{\Gamma_s}{\mu L^2}\left(1 - \Gamma_s\left(D^{-1}\right)_{ss}\right),$$

where the matrix $D$ is given by $D_{mn} = \Gamma_s \delta_{sm}\delta_{sn} + \mu\delta_{nm} + \langle\underline{m}|\mathcal{H}_0 G_0^{-1}\mathcal{H}_0|\underline{n}\rangle$. Since $\Pi$ is a projector one has

$$G_0^{-1} = \frac{1}{\gamma(1 - \Pi) + \mu 1} = \frac{1}{\gamma + \mu}(1 - \Pi) + \frac{1}{\mu}\Pi.$$

Furthermore

$$
\begin{aligned}
\Pi\mathcal{H}_0|n\rangle \otimes |n\rangle & = \Pi\Big(J|n\rangle \otimes |n\rangle - |n\rangle \otimes J|n\rangle\Big) \\
& = \sum_k |k\rangle\langle k| \otimes |k\rangle\langle k|\Big(J|n\rangle \otimes |n\rangle - |n\rangle \otimes J|n\rangle\Big) \\
& = \sum_k \langle k|J|n\rangle\,\delta_{nk}|k\rangle \otimes |k\rangle - \sum_k \langle k|J|n\rangle\,\delta_{nk}|k\rangle \otimes |k\rangle = 0.
\end{aligned}
$$

By using these two identities on finds for the matrix elements $D_{mn}$:

$$
\begin{aligned}
D_{mn} &= \Gamma_s \delta_{sm} \delta_{sn} + \mu \delta_{mn} + \langle m| \otimes \langle m| \mathcal{H}_0 \frac{1}{\gamma(1-\Pi)+\mu\mathbf{1}} \mathcal{H}_0 |n\rangle \otimes |n\rangle \\
&= \Gamma_s \delta_{sm} \delta_{sn} + \mu \delta_{mn} + \frac{1}{\mu+\gamma} \langle m| \otimes \langle m| \mathcal{H}_0 (1-\Pi) \mathcal{H}_0 |n\rangle \otimes |n\rangle \\
&= \Gamma_s \delta_{sm} \delta_{sn} + \mu \delta_{mn} + \frac{2}{\mu+\gamma} \left( \delta_{nm} \langle m| J^2 |n\rangle - \langle m| J |n\rangle^2 \right),
\end{aligned}
$$

where in the last line, we have used the identity:

$$
\begin{aligned}
\langle m| \otimes \langle m| \mathcal{H}_0^2 |n\rangle \otimes |n\rangle &= \langle m| \otimes \langle m| \left( J^2 \otimes 1 + 1 \otimes J^2 - 2J \otimes J \right) |n\rangle \otimes |n\rangle \\
&= 2 \left( \delta_{nm} \langle m| J^2 |n\rangle - \langle m| J |n\rangle^2 \right).
\end{aligned}
$$

Thus we can write the matrix $D$ in the form

$$
D = \Gamma_s |s\rangle \langle s| + \mu \mathbf{1} + K
$$

where $K_{mn} = \frac{2}{\mu+\gamma} \left( \delta_{nm} \langle m| J^2 |n\rangle - \langle m| J |n\rangle^2 \right)$. Now we need to calculate $D^{-1}$. Using the Sherman-Morrison formula one obtains

$$
D^{-1} = \frac{1}{\mu\mathbf{1}+K} - \frac{\Gamma_s}{1 + \Gamma_s \langle s| \frac{1}{\mu\mathbf{1}+K} |s\rangle} \frac{1}{\mu\mathbf{1}+K} |s\rangle \langle s| \frac{1}{\mu\mathbf{1}+K},
$$

and thus for its diagonal matrix elements

$$
\begin{aligned}
\langle s| D^{-1} |s\rangle &= \langle s| \frac{1}{\mu\mathbf{1}+K} |s\rangle - \frac{\Gamma_s}{1 + \Gamma_s \langle s| \frac{1}{\mu\mathbf{1}+K} |s\rangle} \langle s| \frac{1}{\mu\mathbf{1}+K} |s\rangle \langle s| \frac{1}{\mu\mathbf{1}+K} |s\rangle \\
&= \langle s| \frac{1}{\mu\mathbf{1}+K} |s\rangle \left( 1 - \frac{\Gamma_s}{1 + \Gamma_s \langle s| \frac{1}{\mu\mathbf{1}+K} |s\rangle} \langle s| \frac{1}{\mu\mathbf{1}+K} |s\rangle \right) = \frac{\langle s| \frac{1}{\mu\mathbf{1}+K} |s\rangle}{1 + \Gamma_s \langle s| \frac{1}{\mu\mathbf{1}+K} |s\rangle}
\end{aligned}
$$

By substituting this expression into Eq. (44) we finally obtain

$$
\eta \leq \frac{\Gamma_s}{\mu L^2} \left( 1 - \Gamma_s \frac{\langle s| \frac{1}{\mu\mathbf{1}+K} |s\rangle}{1 + \Gamma_s \langle s| \frac{1}{\mu\mathbf{1}+K} |s\rangle} \right) = \frac{\Gamma_s}{\mu L^2} \frac{1}{1 + \Gamma_s \langle s| \frac{1}{\mu+K} |s\rangle} = \eta_{\text{incoh}},
$$

which is expression (45) in the main text.

## C.  1D model with nearest-neighbor hopping

Finally we consider the transport on a 1-dimensional lattice with nearest-neighbor hopping $J$ in detail. The eigenvalues of the hopping matrix $\mathcal{H}_0$ and their corresponding eigenvectors $|k\rangle$ are given by

$$
E_k = 2J \cos\left( \frac{\pi k}{L+1} \right), \qquad k = 1, 2, ...L,
$$

and

$$
\langle k | \alpha \rangle = \sqrt{\frac{2}{L+1}} \sin\left( \frac{\pi \alpha k}{L+1} \right), \qquad \alpha = 1, 2, ...L,
$$

**Explicit expression for $\eta_{\mathrm{coh}}$**

We first want to derive an explicit expression for the upper bound $\eta_{\mathrm{coh}} = 1 - \frac{\mu}{L^2}\sum_{\alpha,\nu}\left(\Omega^{-1}\right)_{\alpha\nu}$ in the low-dephasing limit, eq.(38). The matrix elements of $G$ in the subspace of the null space of matrix $\mathcal{H}_0$ are given by:

$$\Omega_{\alpha\beta} = \left(\langle\alpha|\otimes\langle\alpha|\right)G\left(|\beta\rangle\otimes|\beta\rangle\right)$$

$$= \Gamma_s\,|\langle s\,|\alpha\rangle|^2\,\delta_{\alpha\beta} + \mu\delta_{\alpha\beta} + \gamma\left(\delta_{\alpha\beta} - \sum_k|\langle k\,|\alpha\rangle|^2\,|\langle k\,|\beta\rangle|^2\right)$$

$$= \frac{2\Gamma_s\delta_{\alpha\beta}}{L+1}\sin^2\left(\frac{\pi}{2}\alpha\right) + \mu\delta_{\alpha\beta} + \gamma\left(\delta_{\alpha\beta} - \frac{4}{(L+1)^2}\sum_k\sin^2\left(\frac{\pi k\alpha}{L+1}\right)\sin^2\left(\frac{\pi k\beta}{L+1}\right)\right)$$

$$= B_{\alpha\beta} - \frac{\gamma}{L+1}\left(\frac{1}{2}P_{\alpha\beta} + 1\right),$$

where

$$B_{\alpha\beta} = \left(\frac{2\Gamma_s}{L+1}\sin^2\left(\frac{\pi}{2}\alpha\right) + \frac{\gamma}{2}\frac{2L+1}{L+1} + \mu\right)\delta_{\alpha\beta},$$

and $P_{\alpha\beta}$ are matrix elements of the parity operator:

$$P = \begin{pmatrix} 0 & 0 & . & . & . & 1 \\ 0 & 0 & . & . & 1 & 0 \\ 0 & 0 & . & 1 & 0 & 0 \\ & & & & & \\ 1 & 0 & 0 & 0 & & 0 \end{pmatrix}.$$

In matrix form $\Omega$ reads

$$\Omega = B - \frac{\gamma}{L+1}\left(\frac{1}{2}P + |e\rangle\langle e|\right),$$

where $|e\rangle = (1,1,1,\ldots,1)^\top$. Noting that $\sum_{\alpha,\nu}\left(\Omega^{-1}\right)_{\alpha\nu} = \langle e|\Omega^{-1}|e\rangle$ we can write furthermore

$$\eta_{\mathrm{coh}} = 1 - \frac{\mu}{L^2}\langle e|\Omega^{-1}|e\rangle.$$

In order to calculate the inverse of $\Omega$ we note that the two matrices $P$ and $B - \frac{\gamma}{L+1}|e\rangle\langle e|$ commute, Thus by applying the Sherman-Morrison matrix identity, we can calculate $\Omega^{-1}$:

$$\left[B - \frac{\gamma}{L+1}\left(\frac{1}{2}P + |e\rangle\langle e|\right)\right]^{-1} = \left[B - \frac{\gamma}{L+1}\frac{1}{2}P\right]^{-1} +$$

$$+ \frac{1}{1 - \frac{\gamma}{L+1}\langle e|\left[B - \frac{\gamma}{L+1}\frac{1}{2}P\right]^{-1}|e\rangle}\frac{\gamma}{L+1}\left[B - \frac{\gamma}{L+1}\frac{1}{2}P\right]^{-1}|e\rangle\langle e|\left[B - \frac{\gamma}{L+1}\frac{1}{2}P\right]^{-1}.$$

This then gives

$$\langle e|\Omega^{-1}|e\rangle = \langle e|\left[B - \frac{\gamma}{L+1}\left(\frac{1}{2}P + |e\rangle\langle e|\right)\right]^{-1}|e\rangle = \frac{\langle e|\left[B - \frac{\gamma}{L+1}\frac{1}{2}P\right]^{-1}|e\rangle}{1 - \frac{\gamma}{L+1}\langle e|\left[B - \frac{\gamma}{L+1}\frac{1}{2}P\right]^{-1}|e\rangle}.$$

In the final step we need to evaluate

$$\langle e|\left[B - \frac{\gamma}{L+1}\frac{1}{2}P\right]^{-1}|e\rangle.$$

$B$ is a diagonal matrix whose elements are $B_{\alpha\alpha} = \frac{2\Gamma_s}{L+1} + \frac{\gamma}{2}\frac{2L+1}{L+1} + \mu$ for odd values of $\alpha$ and $B_{\alpha\alpha} = \frac{\gamma}{2}\frac{2L+1}{L+1} + \mu$ for even values of $\alpha$. If we assume an odd number of sites $L$ we find

$$\langle e | \left[ B - \frac{\gamma}{L+1}\frac{1}{2}P \right]^{-1} | e \rangle = \frac{\frac{L+1}{2}}{\frac{2\Gamma_s}{L+1} + \mu + \frac{\gamma}{2}\frac{2L+1}{L+1} - \frac{\gamma}{L+1}\frac{1}{2}} + \frac{\frac{L-1}{2}}{\mu + \frac{\gamma}{2}\frac{2L+1}{L+1} - \frac{\gamma}{L+1}\frac{1}{2}}.$$

From this, we arrive at the result, given in main text:

$$\eta_{\text{coh}} = 1 - \frac{\mu}{L^2}\langle e|\Omega^{-1}|e\rangle = \frac{\Gamma_s}{L}\frac{L\gamma + (L+1)\mu}{2\mu\left(\Gamma_s + \frac{L+1}{2}\mu\right) + \gamma\left(\Gamma_s + L\mu\right)}.$$

**Explicit expression for $\eta_{\text{incoh}}$**

In order to explicitly calculate the upper bound $\eta_{\text{incoh}}$ for the efficiency in the limit of large dephasing, eq.(45), we need to determine $\langle s | \frac{1}{\mu+K} | s \rangle$.

For the 1D model the matrix $K$ takes the following form

$$K = \frac{2J_0^2}{\mu+\gamma}\begin{pmatrix} 1 & -1 & 0 & . & . & 0 \\ -1 & 2 & -1 & 0 & . & . \\ 0 & -1 & 2 & . & & . \\ . & 0 & & . & . & . \\ . & . & & & 2 & -1 \\ 0 & . & . & 0 & -1 & 1 \end{pmatrix}.$$

Hence,

$$\langle s | \frac{1}{\mu+K} | s \rangle = \frac{\det^2(\mu+K_1)}{\det(\mu+K)}$$

where $K_1$ is the $\frac{L-1}{2} \times \frac{L-1}{2}$ matrix

$$K_1 = \frac{2J_0^2}{\mu+\gamma}\begin{pmatrix} 1 & -1 & 0 & & & 0 \\ -1 & 2 & -1 & 0 & & \\ 0 & -1 & 2 & & & \\ & & 0 & & & \\ & & & & 2 & -1 \\ 0 & & & 0 & -1 & 2 \end{pmatrix}.$$

We note that the matrices $K$ and $K_1$ can be rewritten as

$$K = \frac{2J_0^2}{\mu+\gamma}\left(K_0(L) - |1\rangle\langle 1| - |L\rangle\langle L|\right),$$

and

$$K_1 = \frac{2J_0^2}{\mu+\gamma}\left(K_0\left(\frac{L-1}{2}\right) - |1\rangle\langle 1|\right)$$

where $K_0(n)$ is a $n \times n$ matrix of the following form

$$K_0(n) = \begin{pmatrix} 2 & -1 & 0 & . & . & 0 \\ -1 & 2 & -1 & 0 & & . \\ 0 & -1 & 2 & . & . & . \\ . & 0 & . & . & . & 0 \\ . & . & . & . & 2 & -1 \\ 0 & & . & 0 & -1 & 2 \end{pmatrix}.$$

Similarly to the hopping matrix, $K_0(n)$ has a simple diagonal from in momentum space. With this one finds

$$
\det(\mu + K_1) = \det\left[\left(\mu + \frac{2J_0^2}{\mu + \gamma}\left(K_0\left(\frac{L-1}{2}\right) - |1\rangle\langle 1|\right)\right)\right]
$$

$$
= \left(1 - \frac{2J_0^2}{\mu + \gamma}\langle 1|\left(\mu + \frac{2J_0^2}{\mu + \gamma}K_0\left(\frac{L-1}{2}\right)\right)^{-1}|1\rangle\right)\det\left(\mu + \frac{2J_0^2}{\mu + \gamma}K_0\left(\frac{L-1}{2}\right)\right)
$$

$$
= \left(\mu + \frac{4J_0^2}{\mu + \gamma}\right)^{\frac{L-1}{2}}\left(1 - \frac{q}{2}\langle 1|\left(S_{\frac{L-1}{2}}(q)\right)^{-1}|1\rangle\right)\det\left(S_{\frac{L-1}{2}}(q)\right),
$$

where

$$
S_L(q) = \begin{pmatrix}
1 & -\frac{q}{2} & 0 & . & . & 0 \\
-\frac{q}{2} & 1 & -\frac{q}{2} & 0 & & . \\
0 & -\frac{q}{2} & 1 & & & . \\
. & & 0 & & & 0 \\
. & & . & . & 1 & -\frac{q}{2} \\
0 & & . & 0 & -\frac{q}{2} & 1
\end{pmatrix},
$$

and $q = \left(\frac{4J_0^2}{\mu + \gamma}\right)\Big/\left(\mu + \frac{4J_0^2}{\mu + \gamma}\right) < 1$. Similarly we can calculate the second determinant

$$
\det(\mu + K) = \det\left(\mu + \frac{2J_0^2}{\mu + \gamma}\left(K_0(L) - |1\rangle\langle 1| - |L\rangle\langle L|\right)\right)
$$

$$
= \det\left[\left(\mu + \frac{2J_0^2}{\mu + \gamma}K_0(L)\right)\left(1 - \frac{2J_0^2}{\mu + \gamma}\left[\mu + \frac{2J_0^2}{\mu + \gamma}K_0(L)\right]^{-1}\left(|1\rangle\langle 1| + |L\rangle\langle L|\right)\right)\right].
$$

The non-zero eigenvalues of the matrix $\left[\mu + \frac{2J_0^2}{\mu + \gamma}K_0(L)\right]^{-1}\left(|1\rangle\langle 1| + |L\rangle\langle L|\right)$ are

$$
\lambda_{1,2} = \langle 1|\left[\mu + \frac{2J_0^2}{\mu + \gamma}K_0(L)\right]^{-1}|1\rangle \pm \langle 1|\left[\mu + \frac{2J_0^2}{\mu + \gamma}K_0(L)\right]^{-1}|L\rangle.
$$

Here we have used that

$$
\langle 1|\left[\mu + \frac{2J_0^2}{\mu + \gamma}K_0(L)\right]^{-1}|1\rangle = \langle L|\left[\mu + \frac{2J_0^2}{\mu + \gamma}K_0(L)\right]^{-1}|L\rangle
$$

By combining the above results one obtains

$$
\langle s|\frac{1}{\mu\mathbb{1} + K}|s\rangle = \frac{1}{\mu + \frac{4J_0^2}{\mu + \gamma}}\frac{\det^2 S_{\frac{L-1}{2}}(q)}{\det S_L(q)}
$$

$$
\times \frac{\left(1 - \frac{q}{2}\langle 1|\left(S_{\frac{L-1}{2}}(q)\right)^{-1}|1\rangle\right)^2}{\left(1 - \frac{q}{2}\langle 1|(S_L(q))^{-1}|1\rangle\right)^2 - \frac{q^2}{4}\langle 1|(S_L(q))^{-1}|L\rangle^2},
$$

where

$$\langle 1|\left(S_{\frac{L-1}{2}}(q)\right)^{-1}|1\rangle = \frac{\det S_{\frac{L-3}{2}}(q)}{\det S_{\frac{L-1}{2}}(q)},$$

$$\langle 1|(S_L(q))^{-1}|1\rangle = \frac{\det S_{L-1}(q)}{\det S_L(q)},$$

$$\langle 1|(S_L(q))^{-1}|L\rangle = \frac{\left(-\frac{q}{2}\right)^{L-1}}{\det S_L(q)}.$$

We now note that the determinant of $S_n(q)$ can be expressed in terms of the Chebyshev polynomial of the second kind, $U_n\left(\frac{1}{q}\right)$,

$$\det S_n(q) = \left(\frac{q}{2}\right)^n U_n\left(\frac{1}{q}\right),$$

which follows from the determinant identity

$$U_n(x) = \det \begin{pmatrix} 2x & 1 & 0 & . & . & 0 \\ 1 & 2x & 1 & 0 & & . \\ 0 & 1 & 2x & & & . \\ . & 0 & & & & 0 \\ & . & . & & 2x & 1 \\ 0 & & . & 0 & 1 & 2x \end{pmatrix}.$$

The physically relevant case corresponds to a small background absorption rate $\mu$, or equivalently, when $q \approx 1$. Expanding the expression (6) around $q = 1$ yields

$$\det S_n(q) \approx 2^{-n}\left(n + 1 + \frac{1}{3}n\left(n^2 - 1\right)(1 - q) + ...\right)$$

Using the expression for $\det S_n(q)$ in terms of the Chebyshev polynomials one finally arrives at

$$\langle s|\frac{1}{\mu 1 + K}|s\rangle \approx \frac{1}{\mu + \frac{4J_0^2}{\mu + \gamma}}\left(\frac{1}{1 - q}\frac{1}{L} + \frac{L^2 - 1}{6L}\right)$$

$$= \frac{1}{\mu L} + \frac{1}{\mu + \frac{4J_0^2}{\mu + \gamma}}\frac{L^2 - 1}{6L}.$$

This then gives an explicit expression for $\eta_{\text{incoh}}$ presented in the main text of the paper.

**Author contributions** A.S. performed the numerical simulations, R.U. derived the analytic results. The project was conceived and supervised by M.F. All authors discussed the results and contributed to the writeup of the mansucript.

**Funding information** The authors gratefully acknowledge financial support by the DFG through SFB/TR 185, Project No.277625399.

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
