# Peer review of "Dephasing enhanced transport of spin excitations in a two dimensional lossy lattice"

_SciPost Physics_

## Round 1 · Referee Report · Anonymous (Referee 1) · 2025-3-31

Strengths

  1. Interesting and original methodological approach to the problem of energy transport in dissipative hopping lattices. Potential to apply this approach to other model transport systems.
  2. Strong and rigorous analytical results, particularly the bounds on efficiency and transport time and the crucial impact of dephasing in enhancing ( or even allowing) excitations to reach a target site.
  3. Numerical results have been carefully obtained and convincingly compared against analytical predictions.
  4. To the best of my knowledge the discussion and analysis of system size is potentially novel.
  5. Clear and well-presented article, easily accessible to expert and non-expert, alike. The details surrounding the mathematical develop of the numerical monte carlo method and the derivations of the various bounds were particularly welcome.

Weaknesses

  1. The underlying problem and theory of ENAQT ( or noise-assisted dephasing NAT) is now rather old ( 2008 - ) . The lattice system and model studied in this paper is essentially the same as in the first papers looking at ENAQT in the context of exciton transport in photosynthetic proteins. Consequently, the amount of new qualitative insight into ENAQT is somewhat limited.
  2. Introduction focuses a great deal on biological systems, but many relevant citations are missing and some of the presentation of the previous work, particularly w.r.t. this article's contribution, needs to be corrected for the record.
  3. Given the single-particle nature of the dynamics, I was surprised by the small lattice sizes used in the numerical simulations, especially given that previous studies have been able to look at much larger and more complex dynamics.
  4. The main conclusions seem to repeat a number of general facts that have been known/stated in the ENAQT literature many times before. The conclusion concerning critical antenna (lattice) size might be new, but I think this needs to be checked by looking at some of the 'missing' citations, as mentioned above.

Report

This paper presents an interesting approach to energy transport in models that closely resemble those used to understand the physics of 'light-harvesting' systems, and in particular lattice hopping models used to describe photosynthetic chromophore arrays. The Authors employ monte carlo- based methods for the simulation of quantum walks of on an extended 2D lattice in the presence of dephasing and losses, allowing them to explore the role of the lattice size on the efficiency of excitation capture by a target site. The analytical and numerical parts will be of broad interest across physics and chemistry communities interested in this problem, and I believe that this article could be suitable for publication in SciPosts. However, in order to meet all of the relevant acceptance criteria, I believe that significant revision is required. I set these out, and the reasons behind them, in the next section.

Requested changes

  1. The introduction focuses heavily on ENAQT in biological systems such as the FMO complex. Indeed, most of the references relate to studies of this complex, ignoring the vast number of studies on other - typically much larger - complexes such as LHII, LHI, Chlorosomes...given that several of these systems are much closer to the extended 2D lattice considered here, some of these other works should be cited and scanned for any similarities (methods, conclusions concering ENAQT) with the present work. Similarly, an oft-repeated fallacy is that the FMO complex has '100%' efficiency. It can actually be much lower - https://pubs.rsc.org/en/content/articlehtml/2023/cp/d3cp01321a (and refs therein)

As stated, the claims about 'coherence between distant aggregates persist...despite being subjected to noise' isn't true. The Authors should explain carefully what is meant by 'distant', and even by 'coherence ' (do they mean that exciton eigenstates are spread over several monomers, or the coherence between these extended states, as prepared by ultrafast pumping, 'persist'?).

  1. The authors should consider/comment if their results are in any way specific to the highly ordered arrangement of dipoles they use in their model. Real biomolecular aggregates often have misaligned dipoles - would random dipole-dipole (anisotropic) interaction make a difference? Also, given the key conclusions of this article revolve around the lattice size scaling, how does this depend on the range of the dipole interaction w.r.t. the lattice site separations?

  2. The Authors state some standard results for random walks on page 3. Are these quantum or classical walks? Given the existence in this work of both coherent and classical hopping, this is important. Please add clarifications and citations, particularly for the statement about tree-like networks. It should be noted that dissipative quantum walks on bio-like lattices have been studied before, including speed limits and various types of bound. The authors should check their findings against some of this material. For example ( & non-exhaustively):

https://pubs.rsc.org/en/content/articlelanding/2022/cp/d1cp02727a/unauth https://iopscience.iop.org/article/10.1088/1367-2630/12/6/065041/meta https://iopscience.iop.org/article/10.1088/0953-4075/44/18/184012/meta?casa_token=qtWond9LYDcAAAAA:kFyuUxMPjB5SfXpmkGtRuRLD5KUJB61Ryb0ZIS7t_Gu4NrMRJELy9pwZA26Hii-kh-lTkbcFwcctUJbYvIW27WiDLA https://journals.aps.org/prl/abstract/10.1103/PhysRevLett.111.180601

Minor points: Intro mentions 'dark states' with explanation or citations. What are these?

Recommendation

Ask for major revision

---

## Round 1 · Referee Report · Anonymous (Referee 2) · 2025-4-20

Report

The present paper studies the phenomenon of environment-assisted quantum transport (ENAQT), relevant for exciton transport in biological systems, in the case of a 2D square lattice geometry (with periodic and open boundary conditions), in view of possible artificial light-harvesting systems. It considers a single excitation prepared at an initial site, and being collected at a "sink site" at a fixed position. It investigates the behaviour of a transfer efficiency and an average transport time.

ENAQT is characterised by a non-monotonic behaviour of the transfer efficiency as a function of a dephasing noise, with the efficiency increasing for small noise amplitudes, and decreasing for large ones. This phenomenon has been studied in previous works on different geometries.

This work extends it to the case of a 2D square lattice. In this case, it recovers the expected ENAQT behaviour and its explanation in terms of the presence of dark states, and it explains the dark states existence from geometrical considerations. It derives analytic approximations for the efficiency, in the two regimes of small and large dephasing noise, and identifies a characteristic length scale up to which ENAQT is appreciable.

In my opinion, these results are technically correct and valuable, but I have several concerns regarding the paper claims and its presentation, that I would ask the Authors to address before publication in a journal.
In addition, I believe that the paper results do not constitute a significant advance with respect to previous work, and therefore do not meet the acceptance criteria of SciPost Physics, namely of a groundbreaking discovery or a breakthrough on a long-standing problem. I find it instead more appropriate for SciPost Physics Core.

Requested changes

In the following, I will first address the presentation aspects that I found confusing or need improvement, as this might be helpful for the editor as well. These are briefly described here, but more detailed comments are reported as annotations in the attached PDF.

A. Using the symbol $p_{n, m}(t)$ and the name "density matrix" for $ \operatorname{tr} \left{\rho(t) \sigma_m^{\dagger} \sigma_n\right}$ is very confusing in this open system context in which there's also $\rho$ that is also called "density matrix". In particular $p$ is not a probability, and Eq. (27) does not describe a conservation of probability, but of particle density (this initially led me to think that the right-hand-side should have been $0$!). To make these things unambiguous, I would use a different symbol and name.

B. The transfer efficiency (11) and average time (12) are crucial quantities for the paper. Nevertheless, what they actually represent is not sufficiently discussed. For (11), for example, I found the discussion given in Ref. clear [15]: The excitation is trapped with a rate $\Gamma_s$. The probability that the exciton is successfully captured at a target site within the time interval $[t, t + dt]$ is given by $\Gamma_s \langle s |\rho (t)|s \rangle dt$. Thus the transfer efficiency can be described as the probability of trapping during the time interval $[0,t]$ ([15] actually has a $2$, I am not sure why...). Actually, I think the name "trapping probability" would be much clearer than "transfer efficiency", although I understand that the latter is already established. Please find more comments (in particular on the average transfer time) in the PDF attached.

C. Several lengths are defined, such as (6) and (9), but their formulas do not have the dimensions of a length. These should be multiplied by the lattice constant $a$.

D. Overall I found the paper quite long, and often redundant. I think the results on the analytic approximations for the efficiency should come earlier. Instead, several technical aspects could be moved into Appendices. For example the entire section "3.1 Monte-Carlo wavefunction approach" describes a well established approach, and would better be an Appendix.

Apart from these presentation aspects, my main concerns with the claims of the paper are the following.

  1. I find Eq. (27) and its discussion unclear and actually very misleading. This equation assumes that there are no dark states in the problem, thus the excitation prepared is completely dissipated at long times (otherwise the right-hand-side wouldn't be $1$). Nevertheless, this is not true in most cases of interest for the paper, and the predictions of Eq. (27), of $\eta=1$ for $\mu=0$ and $\eta \approx 1$ at small $\mu$, are never realized in any results obtained in the rest of the paper. In fact, for $\mu=0, \gamma=0$, as discussed in 4.1, the efficiency $\eta$ is far from $1$ precisely because several dark states are present. For small $\mu$, $\eta$ is also generically far from $1$, as shown in Fig. 2. For $\mu = 0$ and finite $\gamma$, the Authors report that $\eta=1$ in all cases and therefore this is a bad quantitative measure of transport. It's then not clear to me why to present and discuss such equation at all, but anyway I found the present discussion quite confusing.

  2. One of the main claims of the paper is that "ENAQT disappears for a system size larger than a characteristic value". While the results behind this statement are technically correct, and the corresponding length scale identified seems valuable, I find this statement misleading. In fact, this is a result of how the transfer efficiency $\eta$ is defined, namely by averaging over the site choices where the excitation is initially prepared, which are chosen randomly and uniformly distributed across the entire lattice with linear dimension $L$. As discussed in Fig. 7, for a large $L$ such efficiency is small (decreases with $L$), because most (more) initial sites are just too far from the sink site, and cannot reach it. Instead, stating that the efficiency decreases with $L$ makes one think that this is true keeping the source and sink positions fixed, which is not what the Authors have shown. I understand that the chosen randomly located initial excitation might reflect the physical behaviour of photons collected by an antenna, as commented somewhere in the paper, but all these aspects should be explained more clearly.

  3. An explanation of how the bound (9) (or (34)) arises physically/intuitively is missing.

  4. Following the discussion in point 1., I would also suggest to replace Eq. (28) with $\eta = \eta_{0}(1 - \tau \mu) $, with $\eta_0 = \underset{\mu \rightarrow 0}{\lim} \eta$ and without assuming that the latter is $1$. In addition, this result relating transfer efficiency $\eta$ to the average transport time $\tau$ at small $\mu$ is not sufficiently commented, but seems non-trivial to me. One aspect that I remarked is that $\eta$ is a trapping probability, and as such depends explicitly on the trapping rate $\Gamma_s$, as by Eq. (11). Instead, $\tau$ does not explicitly depend on the trapping rate, as it is better seen if this is re-written as

    $$\tau= \frac{\int_0^{\infty} d t \; t \left\langle\uparrow_s\right| \rho(t)\left|\uparrow_s\right\rangle } {\int_0^{\infty} d t \left\langle\uparrow_s\right| \rho(t)\left|\uparrow_s\right\rangle }$$
    This writing clarifies that this is just an average time, taken over a probability density in time, and the denominator is just its normalization -- in passing, I raiterate that what $\tau$ represents should be better discussed in the paper, and perhaps (12) would be better replaced with expression I wrote here that seems clearer to me. Then, it seems a non-trivial fact that, at small $\mu$, the transfer efficiency $\eta$ is completely determined by the average transport time $\tau$, and that it depends on the trapping rate $\Gamma_s$ only through it, as shown by Eq. (28) (or the one I suggested instead). This indeed shows that the trapping rate is not the limiting factor to the efficiency in this small$-\mu$ regime. (Perhaps surprisingly, this is also true in the Zeno regime of $\Gamma_s \gg J$, for which the small efficiency is also due to a large $\tau$, because hopping into site $s$ is highly inefficient, as shown in Fig. 4.).

Given these considerations, I think the paper needs to be clarified in several ways before publication. In addition, as I said I believe that its results do not meet the acceptance criteria of SciPost Physics, and that it is more appropriate for SciPost Physics Core.

Attachment

Recommendation

Ask for major revision

---

## Editorial Decision

awaiting_resubmission